# Improving Vaccine Assessment Pathways and Decision Making in the Polish Immunization Program

**DOI:** 10.3390/vaccines12030286

**Published:** 2024-03-09

**Authors:** Marcin Czech, Ewa Augustynowicz, Michał Byliniak, Teresa Jackowska, Mikołaj Konstanty, Ernest Kuchar, Agnieszka Mastalerz-Migas, Maciej Niewada, Aneta Nitsch-Osuch, Iwona Paradowska-Stankiewicz, Jarosław Pinkas, Jakub Szulc, Jacek Wysocki

**Affiliations:** 1Department of Pharmacoeconomics, Institute of Mother and Child, 01-211 Warsaw, Poland; 2Business School, Warsaw University of Technology, 02-008 Warsaw, Poland; 3Department of Epidemiology of Infectious Diseases and Surveillance, National Institute of Public Health-National Institute of Hygiene-National Research Institute, 00-791 Warsaw, Poland; 4INFARMA, The Employers’ Union of Innovative Pharmaceutical Companies, 02-670 Warsaw, Poland; 5Department of Pediatrics, Centre of Postgraduate Medical Education, 01-813 Warsaw, Poland; 6Silesian Pharmaceutical Chamber, 40-637 Katowice, Poland; 7Polish Pharmaceutical Chamber, 02-390 Warsaw, Poland; 8Department of Pediatrics with Clinical Assessment Unit, Medical University of Warsaw, 02-091 Warsaw, Poland; 9Polish Society of Family Medicine, 51-141 Wrocław, Poland; 10Department of Family Medicine, Wrocław Medical University, 50-367 Wrocław, Poland; 11Department of Experimental and Clinical Pharmacology, Medical University of Warsaw, 02-091 Warsaw, Poland; 12Department of Social Medicine and Public Health, Medical University of Warsaw, 02-091 Warsaw, Poland; 13School of Public Health, Centre of Postgraduate Medical Education, 01-813 Warsaw, Poland; 14Alab Laboratories, 00-739 Warsaw, Poland; 15Department of Preventive Medicine, Poznań University of Medical Sciences, 60-781 Poznań, Poland

**Keywords:** decision making, immunization policies, national immunization technical advisory groups, Poland, vaccine assessment, medicines policy, health technology assessment

## Abstract

This study examines the vaccine market access pathway in Poland to evaluate its efficiency and propose recommendations for its improvement. The research spans a comprehensive analysis of the vaccine assessment process, ranging from pre-registration to sustainability, encompassing critical components such as national immunization technical advisory groups (NITAGs), health technology assessments, resource evaluations, and decision making. This investigation utilizes a multi-phase approach. Initial desk research aimed to collect accumulated evidence about each step of the vaccine access pathway. This constituted the background for an expert panel discussion (*n* = 13) and a final online questionnaire (*n* = 12), evaluating the timeframes, inclusiveness, transparency, and consistency of the elements of the process. Poland is a late adopter of new vaccines. The country faces budget constraints and lacks a formalized framework for the inclusion of vaccines into the national immunization program. Notably, NITAGs play a crucial role, yet their limited resources and dependence on public health stakeholders diminish their impact. A formal and well-supported advisory body may become a foundation for decision-making processes. The health technology assessment conducted by the national agency is recognized for its timeliness and transparency, though the absence of fiscal analyses in vaccine assessments is identified as a gap that limits the understanding of the value of vaccinations. Resources are key drivers of decision making, and recent changes in legislation offer increased flexibility in financing vaccines. Challenges in the procurement process include a limited consideration of non-acquisition costs and an increased absence of a documented general strategy for immunization program development in Poland, pointing to a need for strategic planning. In conclusion, this study recommends the establishment of a robust NITAG with enhanced resources, incorporating fiscal analyses, transparent resource allocation, and strategic planning for immunization program development. Addressing these recommendations is crucial for optimizing Poland’s vaccine market access pathway, ensuring timely and efficient population-wide vaccine access.

## 1. Introduction

Vaccination programs, their decision-making processes, and their funding greatly vary across Europe. Critical components of national immunization systems are vaccine assessment and governance pathways. The efficient functioning of these pathways improves population access to vaccines and the sustainability of the healthcare system. The increasing complexity of immunization-associated decision making may negatively affect the time to population access to vaccines, which largely varies in Europe [1,2]. Multiple factors and stakeholders shape the immunization landscape in European countries, which differ significantly by immunization budget, country-specific policies, institutions, population access modalities, and funding pathways. Most studies evaluating country policies and decision-making processes focus on Western countries [3,4,5], which allocate more resources to immunization programs than Central and Eastern European (CEE) countries [6]. Public immunization financing data are limited, reporting methods vary [6,7], and a high proportion of total health expenditures are out-of-pocket and private insurance payments [8]. Also, information about the functioning of healthcare systems, the macroeconomic impacts of vaccination programs, and even the disease burden is largely lacking. 

Vaccinations, as public health measures, inherently offer extensive advantages beyond personal benefits, e.g., health- and care-related productivity gains, reductions in disease transmission, herd immunity, and the preservation of antibiotic efficacy. Immunization policies are national competencies aiming to achieve or maintain high vaccination coverage rates. In recent decades, European countries have been working to expand their vaccination coverage rates and offer a broader range of vaccines to target more diseases. This has required strengthening evidence evaluations and decision-making processes. The main areas of development include the increased engagement of country-level national immunization technical advisory groups (NITAGs) [1], health economics, and outcome research [9].

This article aims to evaluate the development of the decision-making process and the financing of immunization programs in Poland, the largest CEE economy. Changes in ongoing immunization policies in the last 20 years are assessed at every step, from the vaccine pre-registration period and vaccine authorization through to expert and health technology assessments, resource evaluations, issuing final decisions on granting public access, procurement, and planning the development of the National Immunization Agenda. An analysis of the current state of the Polish immunization landscape acts as a benchmark for recommendations to improve vaccine assessment pathways and decision making about the immunization program. 

## 2. Materials and Methods

Studied areas of a vaccine market access pathway were previously described by Laige et al., 2021 [2], and include pre-registration, horizon scanning, early advice (three elements of early assessment), marketing authorization, the initiation of assessments, NITAG recommendations, health technology assessments (HTAs), resource assessments, the final decision, inclusion in a national immunization program (NIP), procurement, and sustainability. The study had three phases.

In the first phase, the above elements were evaluated using the desk research method. Related information was extracted from publicly accessible online information sources, the scientific literature, reports, legal acts, and databases available in Polish and English. The analysis of historical data included the period from May 2004 (Poland’s accession to the European Union [EU]) to February 2023. Extracted information was analyzed using quantitative or qualitative descriptive methods.

In the second phase, the advisory panel was assembled, encompassing a blend of current and former stakeholders from the NITAG, representatives from the pharmaceutical chamber and industry, members of various scientific societies, national consultants, stakeholders from HTA bodies, the Ministry of Health (MoH), and the sanitary inspectorate. The majority of these stakeholders were also academic experts. In total, the panel was composed of 12 people. During a series of roundtable meetings (in-person and virtual) between March and October 2023, experts interpreted the results of desk research, sharing their experiences and opinions and discussing each assessment element’s performance, including its timeframes, inclusiveness, transparency, and consistency. Finally, using online anonymous questionnaires evaluating four performance attributes on a 5-point Likert scale (1  =  strongly agree, 2  =  agree, 3  =  neither agree nor disagree, 4  =  disagree and 5  =  strongly disagree), roundtable members quantitatively evaluated elements of the market access pathway. All data are reported descriptively. Agreement and disagreement were responses with a ≤2 and ≥4 ranking, respectively, comprising at least 67% of responses.

In the last phase, based on the results of desk research, quantitative evaluations, and output from the roundtable meetings, experts drew recommendations for improving vaccine assessment pathways and decision making regarding the immunization program.

## 3. Results and Discussion

Poland is a late adopter of new vaccines, with a time to access of > 6 years [2], and is a country in which the budget for vaccinations per capita is low in comparison to other countries with a similar gross domestic product [6,7]. In the period studied, the time for population vaccine access in the NIP was as high as 15 years in the case of a vaccine against human papillomavirus (HPV). Figure 1 illustrates the time to access vaccines within the NIP. This only includes vaccines included in the NIP between 2004 and 2023; however, multiple vaccines still await population access, e.g., 5-in-1 and 6-in-1 multicomponent vaccines, meningococcal MenB and MenACWY vaccines, a tick-borne encephalitis vaccine, and others. The mean time from central approval to availability of recently authorized medicines to patients in Poland evaluated between 2018 and 2021 was 2.26 years [10]. Thus, the time for population vaccine access is >3 times longer (Figure 1) than other medicines.

All 12 advisory panel members responded to an online questionnaire evaluating timeframes, inclusiveness, transparency, and consistency in the vaccine market access pathway, and the results are presented in Figure 2. Table 1 summarizes the key recommendations to improve vaccine assessment pathways and decision making in the immunization program.

### 3.1. Early Steps of Assessment and Vaccine Registration

The early steps of the vaccine assessment process involve pre-registration actions, horizon scanning analyses, and early advice. No public sources were identified regarding any of the three elements of the formal process. However, Polish officials and experts work in immunization taskforces of the European Medicines Agency (EMA), and it cannot be determined whether they convey information to other institutions or the Ministry of Health. The roundtable members confirmed being aware of the future development of vaccines, mainly working with industry partners. This knowledge is provided to public health stakeholders; however, this cannot be considered a formal horizon scanning process. According to the experts, due to the lack of public stakeholder actions in horizon scanning and early advice, the pharmaceutical industry plays a leading role in this area and remains the primary source of information. The exception was the COVID-19 pandemic, when professionals and the public closely followed the development of vaccines; recommendations issued by VT between November 2019 and December 2022 document a complex process of the planning and enrollment of a vaccination program against COVID-19 [11].

Horizon scanning supports planning expenditures to ensure the timely availability of new vaccines [3]; however, countries that implement this process do not necessarily introduce vaccines early [12]. In Poland, the horizon scanning process does not function in other therapy areas where new therapies are emerging, i.e., rare diseases. Due to late access to medical technology in Poland [10], conducting post-registration scanning, which includes reviewing tools and financing models supporting vaccination growth, is more justified. Data on the effectiveness of financing models and the promotion of vaccinations and their population effects mainly come from countries that have introduced vaccinations into their programs. Unfortunately, neither of these analyses are conducted in Poland. Effective methods of immunization are not applied, and sometimes innovative ideas promoting vaccination are restricted. Most experts (>67%) evaluated early steps to vaccine access as noninclusive, not on time, and nontransparent (Figure 2).

The primary stakeholder of vaccine registration in Poland is the Office for Registration of Medicinal Products, Medical Devices and Biocidal Products, and the process itself is central. Its functioning was considered as consistent, inclusive, timely, and transparent by most of the experts (Figure 2). However, registering a vaccine does not mean it is available in Poland; e.g., the shingles vaccine registered in 2018 [13] became commercially available in 2023. Partly, this is due to the lack of pre-registration interest and dialogue between public stakeholders and manufacturers, prioritizing product availability in the early adopting markets. This may be an additional barrier to vaccine market access.

### 3.2. Vaccine Assessments

In Poland, vaccine assessments are initiated by different stakeholders depending on financial sources vaccinations; the MoH initiates the evaluation of vaccines to be used in the NIP and manufacturers are initiators for vaccines to be available in a reimbursement system, whereas territorial governments initiate the assessment of vaccines used in local health programs. From 2023, the National Health Fund has been financing the cost of vaccinations used in the NIP, both compulsory and some recommended vaccines (influenza and HPV); however, before this, compulsory vaccines were funded from the central budget. The process of vaccine inclusion into the NIP does not have a formal framework. Initiation of assessments was one of the most negatively evaluated elements of the vaccine access pathway (Figure 2). Contrary to the reimbursement system and local health programs, the vaccine inclusion process into the NIP has no formal initiator or framework, and, most importantly, no recommendation is binding. The development of the NIP would benefit from the possibility of being initiated by numerous stakeholders, including the NITAG, scientific societies, and patient organizations. Regardless of the initiator, the process should be formalized, structured, and comprehensive, should impose obligations on individual institutions, and should have a time frame.

#### 3.2.1. NITAG Assessments

EU national immunization programs are based on the Immunization Agenda 2030, which recommends prioritizing NITAGs in decision-making processes [14]. Poland has two NITAGs, the VT (established in 2019 based on the former Pediatric Expert Team for the Protective Vaccination Program at the Ministry of Health) [11] and the Sanitary and Epidemiological Council (SEC) [15], which are subsidiaries of the MoH and the Main Sanitary Inspectorate, respectively. In comparison, the SEC was established on the basis of act [16] and the VT was established by the ordinance of the MoH [17]. 

The VT prepares opinions on vaccinations on request of the MoH or the team’s chairman; analyses the vaccination program for consistency with the current epidemiological situation in Poland and worldwide; and reviews the annually issued calendar of the NIP and legislative changes in the field of vaccinations [18]. Public information on the VT’s actions is limited; besides 37 recommendations and statements from the COVID-19 pandemic, the VT’s website [10] contains only one protocol from the first meeting held in August 2019. This document analysis indicates that VT members create recommendations based on the statements of international institutions, the opinions of scientific societies and national consultants, the protocols of the SEC, correspondence between governmental institutions, reports from national surveillance systems, and materials from manufacturers. Earlier recommendations of the VT’s predecessor were documented in the scientific literature [19] and an immunization-related website [20]. However, some of the recommendations have never been published and remain for the use of the MoH only. It was impossible to determine the frequency of VT meetings; however, experts confirmed that after the COVID-19 emergency state, the team’s operations are limited. 

The SEC, an advisory body to the Chief Sanitary Inspector, consists of 15 members, a chairman, and secretaries that are appointed for three years. The SEC works based on internal policy; its scope is broader than only immunization. Minutes from the meetings are not published; however, they can be obtained based on the law on access to public information. In 2017, the anti-vaccine movement obtained and published documents in the article “Secret Documents from Expert Meetings”, supporting conspiracy theories [21]. Based on the documentation and board members’ opinions, SEC meetings were organized from once to thrice per year between 2008 and 2022. Similarly to the VT, the SEC is not working regularly nowadays. 

Polish NITAGs do not have an established role in the healthcare system, and their resources are limited. NITAGs have a low capacity to conduct structured vaccine evaluations because they lack analysts and administrative support. In most cases, NITAGs’ recommendations and statements are not publicly available, which decreases the impact of NITAGs’ work and their transparency. Most importantly, they do not have a binding force. Public health stakeholders are not obliged to consult or consider NITAGs’ recommendations. As a result, the expertise of NITAG members on infectious disease threats and their awareness about the country’s epidemiologic situation, indicating the necessity to adopt an NIP, are not considered. For example, priorities established for developing the NIP and immunization strategy have not been considered not did they result in action, as evidenced by the Supreme Audit Office [22]. There was no consensus on NITAG assessment performance attributes, except for a low transparency (Figure 2). During the COVID-19 pandemic, the NITAGs were highly active; however, evaluating their action before and after is difficult because of a low transparency. Moreover, the perception of NITAG insiders, being members of the panel, may differ from the general view.

In the opinion of the experts, the functioning of several NITAGs in Poland is not rational. During the COVID-19 pandemic, an additional advisory body (COVID-19 Medical Council) subsidiary to the Prime Minister was created. The majority of the members of this group were resigned in January 2022 due to the lack of impact of the recommendations on actual actions. Multiplication of advisory bodies leads to duplication of competencies, restrictions in access to experts, and mismanagement of valuable scientific resources. The lack of the work’s impact demotivates independent initiatives.

The expert panel’s main recommendation was to formally and organizationally establish one NITAG supported by appropriate resources in a model closer to the current functioning of the health technology assessment (HTA) process. Most experts (78%) indicated that an NITAG should be an independent organization with separate structures; however, creating an NITAG in the structures of the Polish Agency for Health Technology Assessment (AOTMiT), supported by a minority of experts (22%), may have some advantages regarding access to analytical and administrative resources, solving conflicts of interest, remuneration of experts working in NITAGs, and synchronously conducting NITAG and HTA assessments. These advantages are outweighed by the necessity of a high level of independence, which is challenging in institutions subsidiary to governmental bodies. NITAGs play a crucial role in the national immunization system, but their development needs time. One of the main challenges is the maintenance of their financial sustainability [1]. A Polish NITAG, except for its advisory role, should be integrated into national immunization decision making to advocate for further investments. In a country like Poland, where decisions about immunization are made at the state level, s strong NITAG may help to integrate vaccinations into disease control programs and primary health services. 

#### 3.2.2. Health Technology Assessment

The AOTMiT, a national HTA body, operates on the basis of an act, and the assessment process has a defined scope and timelines. Experts agreed that the agency’s work is timely and transparent; however, there was no consensus about inclusiveness and consistency (Figure 2). The agency evaluates the vaccines used in the NIP, reimbursed vaccines, and local health programs. Since 2012, the AOTMiT has performed nineteen assessments of vaccines, including four assessments of vaccines intended for use in the NIP (Figure 3) [23]. These four assessments concerned the selection of a vaccine and securing its availability. In most cases, they did not require the submission of separate analyses since manufacturers provided them earlier, e.g., in the reimbursement HTA pathway. The most frequently evaluated primary prophylactic product was a vaccine against influenza. Like other HTA agencies [24,25], the AOTMiT’s vaccine experience is much smaller than drug evaluation. Although there are similarities, the HTA of vaccines differs from the assessment of drugs [9]. This calls for separate evaluation criteria for vaccines, the long-term effects of population immunization programs, and both assessments and ongoing evaluations of effects based on the data from the surveillance system. From January 2030, all vaccines will be included in the Joint Clinical Assessment; however, the final shape of this assessment needs to be established [26]. 

New types of assessment require efficient surveillance systems. There are considerable regional differences in the sensitivity of infectious disease surveillance in Poland [21,22,23,24] and between European countries [12]. An underestimated incidence of diseases may hinder accurate understanding and response. Inaccurate epidemiologic estimates limit the perception of vaccine value in assessing possible benefits and forecasting, impair monitoring of vaccine impact, delay detection of responses, and lead to difficulties in identifying emerging strains and vaccine mismatch. All of these result in a lack of evidence for policymaking. The COVID-19 pandemic significantly burdened national surveillance systems [25]. A robust laboratory infrastructure, accurate diagnostic testing, and reporting by healthcare professionals are necessary elements of an efficient surveillance system, providing evidence to support immunization decisions.

The COVID-19 pandemic has shown how strong an impact infectious diseases can have on the economy [27]. This highlighted the importance of economic and fiscal analyses in the assessment of vaccines. In most cases, vaccinations meet the cost-effectiveness requirement; the health gains are worth the cost compared to alternative treatment options. However, vaccinations found to be cost-effective are frequently unaffordable due to budget constraints. Nowadays, analysis of the economic and fiscal impact of vaccines, or more broadly, vaccination programs, is not carried out in Poland and has no legislative basis. Fiscal modeling frameworks can inform how vaccinations affect public accounts (e.g., taxes and social transfers) and support setting healthcare priorities, including the timing and funding streams of allocation of resources [25,28,29,30]. This could be the task of a properly prepared and equipped NITAG. According to the International Society for Pharmacoeconomics and Outcomes Research, institutions responsible for developing economic analyses for novel vaccination initiatives should consider the country’s policy goals and the decision-making context, i.e., the rate of vaccine acceptance in the target population, the effect of the vaccination program on the incidence of the disease over time, the costs of executing and maintaining the vaccination program, and the changes to expenses and health outcomes related to a targeted disease [31]. Broader analyses considering the achievement of the aims of the vaccination program from a time perspective may also change the perception of the value of immunization. 

#### 3.2.3. Assessment of Resources

A limited budget was the most important reason not to include new vaccinations in the NIP [2,22]. Until 2022, the NIP calendar’s model of financing compulsory vaccinations was determined each year in a national-budget-related act. In 2012–2016, the MoH requested finance from the Ministry of Finance multiple times and informed the Council of Ministers about the necessity of increasing the financing of vaccinations. However, the MoH declared support for introducing new vaccines into the NIP and said that covering related costs with allocated funds was impossible [22]. Nowadays, the National Health Fund’s budget covers vaccinations in the NIP within a part dedicated to medicines, up to 17% of the total budget. This limits the maximum value of the payer’s spending. The total reimbursement budget increases with the amount allocated for healthcare financing, which is aimed to be 7% of gross domestic product. The recent amendment to the Reimbursement Act increased the MoH’s flexibility in financing vaccines and vaccinations from different sources governed by a range of acts and administrative regulations. In addition to the resources of the National Health Fund, vaccinations can be financed from the medical fund established by the President of Poland and the MoH [32]. For example, vaccinations against HPV are funded from this source for adolescents (12 and 13 years old), whereas all other adolescents and adults receive vaccinations financed from sources in the National Health Fund. 

One of the most important pillars in Poland’s healthcare development strategy is reversing the healthcare services pyramid, aiming to decrease secondary and tertiary service spending. However, the strategic document [33] does not mention vaccinations; prophylactics, diagnostics, and early access to primary care are fundamental to the planned change. The roundtable experts found that recent changes in the Reimbursement Act increased the MoH’s independence in allocating resources and the number of possible sources of financing primary prophylactics supportive of the development of immunization in Poland. However, it is unknown how resources will be allocated. The experts agree that resource allocation is inconsistent and lacks transparency and consistency (Figure 2); i.e., despite an existing consensus and the MoH’s support of changes in the NIP, vaccines have been waiting to be included in the schedule for many years. For example, Poland is the only country in the EU still using whole-cell pertussis vaccines in primary immunization. Multiple vaccines are recommended in the schedule without or with limited public funding [6,34]. However, evidence was generated but not neglected; it frequently did not lead to decision making due to a lack of resources. This leads to inequities in vaccine access. 

### 3.3. Decision making

The MoH’s decisions on making vaccines available are inextricably linked with their financing/reimbursement from public funds. Positive decisions regarding population access to vaccines are communicated publicly in the reimbursement list (financing to eligible persons) or the NIP calendar published annually by the Main Sanitary Inspectorate [34]. In the future, they will be announced as an ordinance to the MoH since the Constitutional Court in 2023 questioned their forms of communication. Thus, the NIP will require a broader consultation than only technical communications. This may increase the transparency of the development of the NIP. The argument regarding the lack of decisions can be found in the documents of the Supreme Audit Office and in the MoH’s answers to questions from Members of Parliament. A lack of resources was the main reason for delays in population access to vaccines. 

Resource assessments are of key importance for decisions. Unification and concentration of vaccination funding in the National Health Fund may result in decision independence and accelerate NIP development. 

Recently, some vaccines have been listed together with drugs that are 100% reimbursed for children, adolescents < 18 years old, and people aged ≥ 65 years old. This is an additional form of financing a primary prevention for people in age-related risk groups. 

Territorial governments, employers, and patients supplement the central governmental financing system. Interestingly, contrary to most European countries, where territorial governments are responsible for implementing vaccination programs, in Poland, they both plan and perform them upon a favorable decision of the AOTMiT. Experts recognize these activities as often better planned and more effective than central ones. However, their local character leads to disparities in access to healthcare in the country.

### 3.4. Procurement

Public procurement tenders are the basis for purchasing vaccines to implement the NIP and other primary prevention programs. The selection of the most advantageous offer is based on the criteria, including the tender’s essential terms. As many as 73% of tenders in 2011–2015 had price as the only criterion for evaluating the offer. The product’s quality and composition are secondary criteria or not included in tenders. This results in limited access to multicomponent vaccines in the NIP. In the expert panel’s opinion, in most cases, the tender process does not account for non-acquisition costs, such as storage, transport, and administration. Moreover, it does not account for patients’ preferences; most parents prefer multicomponent vaccines for mandatory vaccinations. Consequently, around 60% of parents vaccinate their children with privately purchased vaccines, financing compulsory vaccinations guaranteed by the state [22].

Competitive, sole-source procurement dominates in Poland. These are one-time interactions between the payer and a manufacturer that involve the delivery of vaccines in a specific quantity at a given time. Between 2011 and 2015, 11% of tenders were canceled due to a lack of offers. This stresses the importance of balancing the capabilities and interests of suppliers and the state. Most recently, a risk-sharing mechanism was used to ensure the launch of a vaccination program against human papillomavirus, the first fully financed recommended vaccination. Simultaneously, not one but two manufacturers delivered products for the vaccination campaign. This, as well as the participation in the centralized procurement of COVID-19 vaccines, was an exception to the rule. A broader use of risk-sharing mechanisms would increase the price flexibility of manufacturers and enable the financing of more vaccines than before. In addition, purchasing from more than one manufacturer can improve the country’s vaccine security.

### 3.5. Immunization Program Development

Contrary to other countries, Poland does not have a documented general strategy for the development of an immunization program. As mentioned before, despite the alignment of many strategies with local vaccination policies, the current strategic health framework (2021–2027 and beyond) does not mention vaccinations [33]. The former perspective (2018–2022) was primarily focused on primary prevention and led to several improvements, including broadening vaccine access in the NIP and healthcare professionals’ vaccination competencies [35].

Specifically, for certain vaccines, only vaccinations against COVID-19 and HPV were part of broader strategies [36,37]. The plan for population access to the HPV vaccine was included in the National Oncological Strategy, approved by Parliament in 2019. In line with this strategy, a vaccination program against HPV was planned in 2021 for adolescent girls and boys in the following years [37]. However, vaccinations for children aged 12 and 13 started in mid-2023 (Figure 1) [38]. The strategic aim is to vaccinate 60% of adolescents against HPV by the end of 2028 [37]. The program’s early performance is around 10%, and most patients (90%) were vaccinated with the nine-valent product (data from 1 September to 23 August). To increase coverage, the two-valent vaccine was 100% and 50% reimbursed for children and adolescents <18 years old and adults, respectively. However, the strategy quickly adapted to the program’s efficiency and vaccine demand, but it did not benefit from the experiences of other countries that implemented the program previously; i.e., it is not supported by the outcomes of post-registration scanning [39].

Poland has started to face challenges related to population aging, and the recent changes in the reimbursement rules, i.e., financing recommended vaccinations, including some vaccines on lists of free medicines, may support the development of a lifetime vaccination strategy. The evaluation of the perspectives on immunization strategies was inconclusive (Figure 2). Having developed a pediatric immunization strategy, Poland must align with the current Immunization Agenda [12], implementing an adult vaccination program or a broader long-term vaccination program to fill the risk gaps associated with an aging society. A national immunization strategy forms the basis for the assessment of vaccines, which requires accounting for policy objectives in a country-specific context [31].
vaccines-12-00286-t001_Table 1Table 1Recommendations for improvements to vaccine market access and decision process pathways in Poland.StepRecommendationsPre-registrationAs Poland remains a late adopter of new vaccines, a post-registration scanning process should be carried out to review tools and financing models supporting the increase in vaccination rates.Later, Poland should introduce a regular horizon scanning process to identify technologies in development or join related international activities.Initiation of assessmentDevelop a formal route for initiating vaccine evaluation within the NIP, specifying initiators, assessment elements, institutions responsible for conducting the assessments, and process timeframes.Set vaccination coverage goals and timeframes for achieving them.NITAG assessmentSet up an independent NITAG, empowered and strengthened with administrative and analytical resources responsible for:
○Pre-registration activities (horizon scanning).○Post-registration activities (reviews of financing and organizational models supporting vaccination coverage).○Initiating vaccine assessments to make recommendations for the appropriateness of funding and achievable goals.○Assessments of the broad economic and fiscal impact of vaccination programs in terms of prospectively set goals and timeframes for achieving them [30].Establish the course for streamlining and advancing the vaccination program for children and adults.HTABroaden the traditional assessment of effectiveness, safety, and cost-effectiveness toward a comprehensive assessment of the costs and benefits associated with achieving vaccination program objectives in an appropriate time horizon.Assess the impact on the budget, assuming the implementation of program objectives from a time perspective.Shorten assessment times to enable early access to new vaccines.Resource assessment and allocationVaccination prioritization (time frame and expenditure) based on health goals and HTA and NITAG analyses.Make the system more flexible, financing vaccinations in a similar way to financing medicines.Support vaccine uptake and vaccination rates by introducing a payment model that favors free-of-charge reimbursement or a nominal patient co-pay.ProcurementMulti-winner tenders to increase safe access/supply and patients’ individual preferences.Widen the use of risk-sharing mechanisms.Communicate and plan tenders in advance.Fair and transparent criteria driving tender decisions.Immunization program developmentAlignment with the current Immunization Agenda [13] and focus on adult immunization.HTA, health technology assessment; NIP, national immunization program; NITAG, national immunization technical advisory group.


### 3.6. Limitations of the Study

This study has some limitations. An inherent limitation of expert’s opinion is the subjectivity of panel members composed of people with different professional backgrounds and experiences, which affects their opinions, views, and responses. Although the quantitative evaluation method aimed to reduce the influence of dominant individuals through anonymous responses and high response rates, the voting was preceded by a desk research data review and discussion, potentially affecting the final responses. The study did not aim to reach the pre-defined threshold of consensus about selected performance indicators of each step of vaccine access pathways through multiple rounds of voting, but rather to capture opinions based on the collected evidence and discussion.

## 4. Conclusions

This study sheds light on the intricacies of Poland’s vaccine market access pathway, revealing both strengths and challenges. The analysis, spanning pre-registration to sustainability, underscores the need for targeted interventions to fortify the country’s immunization landscape. 

We believe that the role of NITAGs in Poland’s vaccine decision-making process can be pivotal, yet limited resources and a lack of formalization hamper their impact. International examples from countries with robust immunization systems emphasize the need for well-established, independent NITAGs. For instance, models like those in the United Kingdom, Germany, and France showcase the efficacy of independent bodies that provide evidence-based recommendations with binding authority [1,2,5]. Poland should consider restructuring its NITAG, providing sufficient resources, independence, and formal authority, mirroring successful models worldwide.

Poland’s health technology assessment (HTA) process is commended for its timeliness and transparency. However, the absence of economic and fiscal analyses, the long-term impact of vaccines on indirect cost, and the compatibility with epidemiology surveillance pose significant problems. Multiple countries have successfully incorporated economic evaluations into vaccine assessments, providing a comprehensive view of immunization programs’ cost-effectiveness and gaining a governmental perspective of vaccination’s value [28,40,41,42]. Poland should consider integrating robust economic analyses within its NITAG and HTA frameworks to inform decisions on resource allocation, ensuring that cost-effective vaccines are accessible within budgetary constraints.

The recent legislative changes in resource allocation offer increased flexibility for the MoH, yet challenges may persist. European countries have stable or dynamic budgets, supporting vaccinations with transparent resource allocation mechanisms and strategic immunization planning [3]. Also, in Poland, sustained resource commitment with transparent accountability mechanisms is required. This requires a strategic approach, documenting a comprehensive immunization strategy considering pre-registration to post-implementation aims. This strategy should align with broader healthcare goals and promote efficient resource utilization compatible with epidemiology surveillance. 

The procurement challenges identified in this study, such as limited considerations of non-acquisition costs and patients’ preferences, underscore the importance of a reformed approach. Countries like Germany, Italy, and Sweden have exemplary transparent and patient-centric procurement strategies, where non-price factors and patient preferences play a significant role [43]. Poland should consider revising its procurement criteria to ensure a holistic evaluation, encompassing factors beyond cost, such as storage, transport, and patient preferences. Exploring risk-sharing mechanisms could also enhance price flexibility and increase vaccine security [44].

The absence of a documented general strategy for immunization program development in Poland necessitates strategic planning. Countries may exemplify the benefits of overarching immunization strategies. Poland should develop a comprehensive country-specific strategy that outlines long-term goals, incorporates lessons from successful programs, and adapts to changing epidemiological landscapes.

In conclusion, this study provides a nuanced understanding of Poland’s vaccine market access pathway, pointing to areas for improvement. Poland has the opportunity to fortify its immunization system, ensuring timely and equitable access to vaccines for its population. The proposed solutions align with global best practices, offering a roadmap for Poland to enhance its vaccine decision-making processes and contribute to the broader global health agenda.

## Figures and Tables

**Figure 1 vaccines-12-00286-f001:**
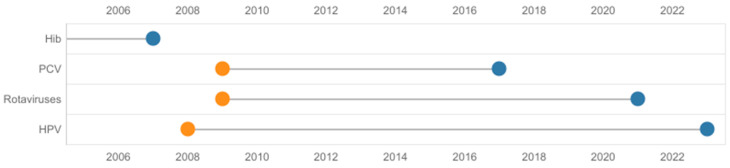
Time from registration to population access of vaccines financed from 2004 to 2023 in Poland. Information based on the dates of first registration of vaccines (orange) and financing from public sources (blue) (national immunization program or medical fund). *Haemophilus influenzae* type b vaccine was registered for the first time in 1995. Hib, *Haemophilus influenzae* type b; HPV, human papillomavirus; PCV, pneumococcal conjugate vaccine.

**Figure 2 vaccines-12-00286-f002:**
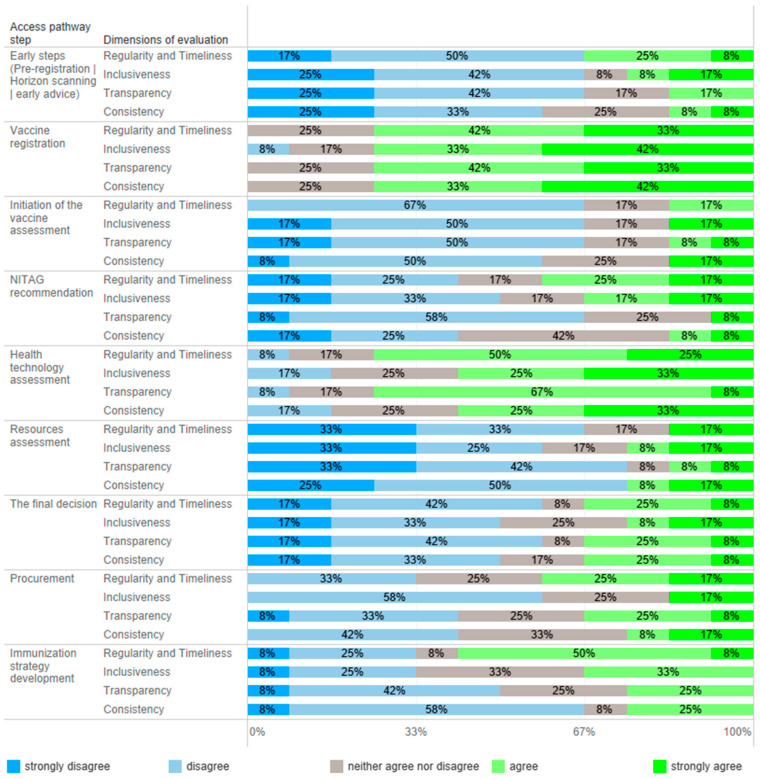
Quantitative evaluation of vaccine market access steps in Poland. Experts (*n* = 12) were asked to determine the level of agreement with statements about regularity/timeliness, inclusiveness, transparency, and consistency in every step of the market access pathway.

**Figure 3 vaccines-12-00286-f003:**
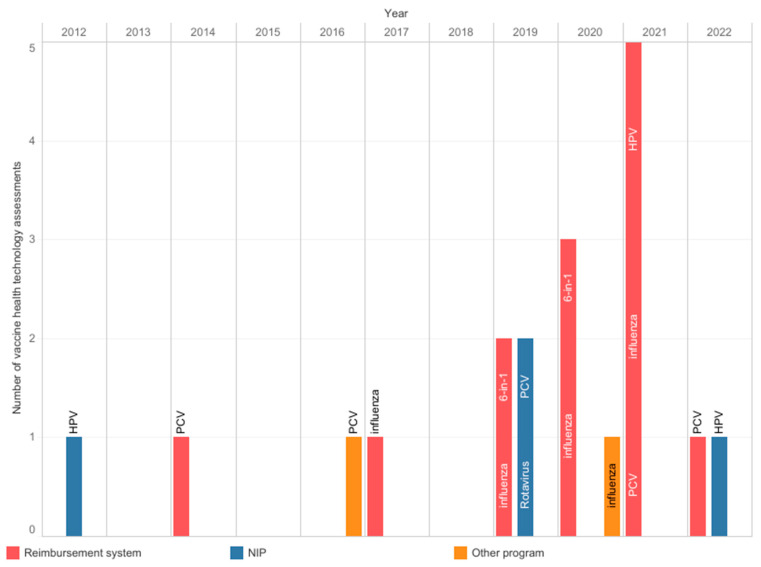
Health technology assessments were performed at the Agency for Health Technology Assessment and Tariff System from 2012 to 2022. Information from archives of the Public Information Bulletin of the Agency [23]. 6-in-1: multicomponent diphtheria, tetanus, pertussis, polio, *Haemophilus influenzae* type b, and hepatitis B vaccine; HPV: human papillomavirus; NIP: national immunization program; PCV: pneumococcal conjugate vaccine.

## Data Availability

Requests for data generated in the study can be forwarded to the corresponding author. Data-sharing requests will be considered on a case-by-case basis.

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
