# Peer review of "Improving Vaccine Assessment Pathways and Decision Making in the Polish Immunization Program"

_vaccines, 2024, doi:10.3390/vaccines12030286_

Round 1
Reviewer 1 Report
Comments and Suggestions for Authors
The paper deals with a crucial public health issue. The paper is well written and authors have have in deep analyzed the theme. I suggest to accept without changes
Author Response
Answer to review of the manuscript entitled Improving vaccine assessment pathways and decision-making in the Polish immunization program (vaccines-2851443).
Reviewer 1
The paper deals with a crucial public health issue. The paper is well written and authors have have in deep analyzed the theme. I suggest to accept without changes.
Thank you for such a good evaluation. However, the remaining reviewers asked for some corrections, which were introduced and are tracked in the manuscript.
Sincerely yours,
Reviewer 2 Report
Comments and Suggestions for Authors
The Article
Improving vaccine assessment pathways and decision-making in the Polish immunization program has been reviewed.
Vaccine market access pathways across are complex, lengthy, and heterogeneous, and the knowledge base to inform recommendations for optimization is the corner stone to improvement. The authors conducted a comprehensive evaluation of vaccine assessment pathways in two phases which include a desk research phase and a panel of expert interviews to identify barriers, drivers, and recommendations. They concluded that there is significant potential for improvement in different aspects mainly because of lack of resources. In response policies should be implemented in Poland to improve access to all vaccines, including routine ones, and form the foundation upon which a consistent vaccine ecosystem can be created.
The work is solid yet there are a few typo errors to be corrected :
Abstract:
line33 –Stakeholder should be in plural , I assume that there are more than one in the public health system.
Line 178 3.2 change Vaccines assessments to Vaccine assessments
Line 360-364
Consider splitting, the phrase is too long.
Table 1 change title to: Recommendations for improvements….
The results and discussion part is too long and becomes repetitive. No limitations to the study have been included.
Conclusions are also too long
Comments on the Quality of English Language
The work is solid yet there are a few typo errors to be corrected :
Abstract:
line33 –Stakeholder should be in plural , I assume that there are more than one in the public health system.
Line 178 3.2 change Vaccines assessments to Vaccine assessments
Line 360-364
Consider splitting, the phrase is too long.
Author Response
Answer to review of the manuscript entitled Improving vaccine assessment pathways and decision-making in the Polish immunization program (vaccines-2851443).
Reviewer 2
Thank you for the review. We performed part of the requested changes. We agree with your queries; however, elements like combining results and discussion and extending conclusions were performed on purpose. We provide a more detailed explanation below.
Vaccine market access pathways across are complex, lengthy, and heterogeneous, and the knowledge base to inform recommendations for optimization is the corner stone to improvement. The authors conducted a comprehensive evaluation of vaccine assessment pathways in two phases which include a desk research phase and a panel of expert interviews to identify barriers, drivers, and recommendations. They concluded that there is significant potential for improvement in different aspects mainly because of lack of resources. In response policies should be implemented in Poland to improve access to all vaccines, including routine ones, and form the foundation upon which a consistent vaccine ecosystem can be created.
The work is solid yet there are a few typo errors to be corrected :
Abstract:
line33 –Stakeholder should be in plural , I assume that there are more than one in the public health system.
Line 178 3.2 change Vaccines assessments to Vaccine assessments
Line 360-364 Consider splitting, the phrase is too long.
Table 1 change title to: Recommendations for improvements….
The results and discussion part is too long and becomes repetitive. No limitations to the study have been included.
Conclusions are also too long.
Thank you for your comments. We corrected the typo errors indicated. We agree that in some parts of the manuscript, there are repetitions; however, early steps of the vaccine access path influence the latter and the aim to retain these contexts. Before the submission, we agreed to the non-standard format of the manuscript, which includes results and discussion combined and extended conclusions. It was to avoid separate reporting of outcomes of desk research discussion and vaccine access pathway evaluation, which would result in even more repetitions.
We include section 3.6. Limitations of the study outlining limitations of the method based on expert consensus and possible bias coming from a heterogeneity of the panel. All changes are in track changes mode.
Sincerely,

Reviewer 3 Report
Comments and Suggestions for Authors
This is an excellent, well researched and written article on an important topic, delays in vaccine approval and use in Poland.
Results
“In the period studied, the 118 time for populational vaccine access in the NIP reached even 15 years in the case of a 119 vaccine against human papillomavirus (HPV). Figure 1 illustrates the time to access vac- 120 cines within the NIP. This includes only vaccines included in the NIP between 2004 and 121 2023; however, multiple vaccines still await populational access, e.g., 5-in-1 and 6-in-1 122 multicomponent vaccines, meningococcal MenB and MenACWY vaccines, tick-borne en- 123 cephalitis vaccine, and others. The mean time from central approval to availability of re- 124 cently authorized medicines to patients in Poland evaluated between 2018 and 2021 was 125 2.26 years [10]. Thus, the time for populational vaccine access is >3 times longer (Figure 1) 126 than other medicines.”
[this is your key finding. Can you discover why the process is so lengthy and longer than for medicines? Is it because patients or patient advocacy groups or physicians advocate/demand medicines?]
Two sets of data would immeasurably strengthen your article:
1. Identify the rates of vaccine-preventable diseases in Poland compared to countries which have high rates of use of relevant vaccines
2. Identify countries with faster rates of vaccine preventable diseases and compare their mechanisms of adoption to those in Poland. Are there particular steps in Poland which entail long delays (e.g. approval by particular committees or ministers?) These data could explain to the key decision makers the high costs of delay in approvals resulting in unnecessary illnesses and use of the health care system.
English text is excellent. A few typos:
“fundament of decision-making processes”.
[change to foundation for decision-making processes]
53 “to populational ‘” [change to population] [change throughout manuscript please to population; populational does not exist as an adjective]
“high proportion of total health expenditures are out-of-pocket and private insurance payments”
[change are to is. The subject is proportion, which is a singular subject]
Reviewer 4 Report
Comments and Suggestions for Authors
The paper is well-written and interesting, the paper examines the vaccine market access pathway in Poland to evaluate its efficiency and propose recommendations for improvement. The investigation utilizes a multi-phase approach, employing desk research, expert panel discussions, and online questionnaires to gather stakeholder insights. However, there are still some minor revisions that need to be done. If they can be done, the manuscript could make more contribution. After a careful review of this manuscript, the reviewer has the following comments:
1. The abstract should briefly describe the research process, research object, number of samples and results. I suggest that the authors should add it.
2. I suggest the author can cite more journal papers in the past three years.
3. What is the main question addressed by the research? The author could have stated more clearly in Introduction section.
4. This article does not have the “4.Results” section. It is suggested that the author should adjust the content of the manuscript.
Comments on the Quality of English LanguageMinor editing of English language required
Round 2
Reviewer 3 Report
Comments and Suggestions for Authors
The authors have made all the changes I requested and the manuscript is suitable for publication based on the authors' replies.